# Beyond the revised cardiac risk index: Validation of the hospital frailty risk score in non-cardiac surgery

**Pishoy Gouda[1], Xiaoming Wang[2], Erik Youngson[2], Michael McGillion[3], Mamas A. Mamas[4], Michelle M. Graham[1] ***

**1** University of Alberta, Division of Cardiology and Mazankowski Alberta Heart Institute, Edmonton, Alberta, Canada, **2** Research Facilitation, Alberta Health Services, Edmonton, Alberta, Canada, **3** School of Nursing, Faculty of Health Sciences and Population Health Research Institute, McMaster University, Hamilton, Ontario, Canada, **4** Keele Cardiovascular Research Group, Centre for Prognosis Research, Keele University, Keele, Newcastle, United Kingdom

* mmg2@ualberta.ca

**Data Availability Statement:** The data underlying this article was provided by the Government of Alberta under the terms of a research agreement. The authors had no special access privileges to the

## Abstract

Frailty is an established risk factor for adverse outcomes following non-cardiac surgery. The Hospital Frailty Risk Score (HFRS) is a recently described frailty assessment tool that harnesses administrative data and is composed of 109 International Classification of Disease variables. We aimed to examine the incremental prognostic utility of the HFRS in a generalizable surgical population. Using linked administrative databases, a retrospective cohort of patients admitted for non-cardiac surgery between October 1st, 2008 and September 30th, 2019 in Alberta, Canada was created. Our primary outcome was a composite of death, myocardial infarction or cardiac arrest at 30-days. Multivariable logistic regression was undertaken to assess the impact of HFRS on outcomes after adjusting for age, sex, components of the Charlson Comorbidity Index (CCI), Revised Cardiac Risk Index (RCRI) and peri-operative biomarkers. The final cohort consisted of 712,808 non-cardiac surgeries, of which 55·1% were female and the average age was 53·4 +/- 22·4 years. Using the HFRS, 86.3% were considered low risk, 10·7% were considered intermediate risk and 3·1% were considered high risk for frailty. Intermediate and high HFRS scores were associated with increased risk of the primary outcome with an adjusted odds ratio of 1·61 (95% CI 1·50–1·74) and 1·55 (95% CI 1·38–1·73). Intermediate and high HFRS were also associated with increased adjusted odds of prolonged hospital stay, in-hospital mortality, and 1-year mortality. The HFRS is a minimally onerous frailty assessment tool that can complement perioperative risk stratification in identifying patients at high risk of short- and long-term adverse events.

## Introduction

Pre-operative frailty is associated with short- and long-term adverse outcomes [1–3]. However, there is no gold standard for defining frailty, which most commonly is described as a clinically recognizable state of increased vulnerability due to aging-associated decline in reserve and

data. Inquiries respecting access to the data can be made, as the authors did, by contacting health. resdata@gov.ab.ca. This study is based in part on data provided by Alberta Health and Alberta Health Services. We thank the Customer Relationship Management and Data Access Unit at Alberta Health for creating the linked database. The interpretation and conclusions are those of the researchers and do not represent the views of the Government of Alberta. Neither the Government of Alberta not Alberta Health express any opinion in relation to this study.

**Funding:** The author(s) received no specific funding for this work.

**Competing interests:** The authors have declared that no competing interests exist.

function across multiple physiological domains [4]. While there are numerous frailty scores that accurately identify and risk stratify frail individuals, these tools can be time consuming and not easily applied at the bedside [5]. Furthermore, such frailty scores cannot be applied retrospectively in the development of prognosis tools, use for risk stratification and benchmarking of services. Despite these challenges, many key stakeholders advocate for wide-scale implementation of frailty assessment. The benefits of early identification of frailty include preventing/slowing further decline, prognostication, avoidance of inappropriately aggressive therapies and encouraging goals of care discussions [6]. Currently the gold standard of risk stratification in pre-operative settings include the use of the Revised Cardiac Risk Index (RCRI) and pre-operative B-type natriuretic peptides (BNP) [7, 8]. However, frailty is not captured or taken into account using these measures. Recently the use of administrative databases has been targeted as a convenient, cost-effective, automated method of identifying frailty. Gilbert et al. developed and validated the Hospital Frailty Risk Score (HFRS) in medical inpatients, which is composed of 109 International Classification of Disease (ICD-10) diagnostic codes to create a numerical score [9]. This score was demonstrated to have fair to moderate overlap with previously validated clinical frailty scores such as the Fried Phenotype and Rockwood Frailty Index and identified high frailty risk patients that had a significantly higher 30-day mortality and readmission rates.

We assessed the validity of the HFRS in a large perioperative cohort in the peri-operative period to predict adverse outcomes.

## Methods

### Study design

A retrospective cohort of all patients selected elective non-cardiac surgery (S1 Table) between October 1st, 2008 and September 30th, 2019 in Alberta, Canada, was created using linked administrative databases to identify 798,918 admissions for non-cardiac surgery. Surgeries were excluded if they occurred within one-year of the index surgery (n = 86,110; i.e. only the first surgery for each patient within the time period was included), with a final cohort of 712,808 non-cardiac surgeries. The following databases were linked using individual patient provincial health numbers: 1) Alberta Inpatient Discharge Abstract Database (DAD) that includes information on all admissions to acute care facilities including most responsible admission diagnosis (coded using International Classification of Diseases, Canadian Enhancement; ICD-10) and in-patient surgical procedures (coded using Canadian Classification of Health Intervention codes); 2) The Pharmaceutical Information Network (PIN) Database that captures outpatient medication dispensations in Alberta, regardless of medication insurance coverage. Over the counter medications are not captured in this database; 3) The Alberta Health Care Practioner Claims Database, which identifies physician billing claims; 4) Alberta Provincial Registry data which ascertains start and stop dates of Alberta Health Care Insurance Plan coverage, including dates of death; 5) Alberta Health Services (AHS) Laboratory Database, which is the repository for all in-patient and out-patient laboratory investigations.

### Hospital frailty risk score and RCRI score calculation

Using the methodology previously described by Gilbert et al. [9], the Hospital Frailty Risk Score was calculated using 109 ICD codes identified in any position. (S2 Table) with variable weighting to calculate the Hospital Frailty Risk Score [9]. Eligible ICD codes included those within 2 years of surgery date. A score of $< 5$ indicated low risk, 5–15 intermediate risk and $>15$ high risk.

Our method for calculating the RCRI score using administrative data has previously been reported [10]. Briefly, the RCRI was calculated by 1-point assignments for the presence of each of the following variables: 1) history of ischemic heart disease 2) heart failure 3) stroke or transient ischemic attack 4) insulin treated diabetes 5) creatinine $\geq$ 177umol/L and 6) high-risk surgery (intra-thoracic, vascular and intra-peritoneal), for a maximum score of 6. These data were extrapolated from the Discharge Abstract Database using the International Classification of Disease (ICD) codes. ICD-10 codes used included I20—I25 for ischemic heart disease; I50 for heart failure, I60—I69 for cerebrovascular disease; E10-14 for diabetes. Insulin use was determined from the PIN Database within 100 days prior to surgery. The most recent pre-operative creatinine, up to 91 days prior to surgery, was obtained from the AHS Laboratory database.

## Outcomes

Our primary outcome was a composite of death, myocardial infarction or cardiac arrest at or before 30-days. Other short-term outcomes included: prolonged hospital stay, in-hospital mortality, 30-day mortality, hospital readmission for any reason or emergency department presentation for any reason at 30-days. A prolonged hospital stay was defined as greater than expected length of stay for that particular surgery plus one day. Expected length of stay for surgery type was based on institutional standards (S3 Table). Long-term outcomes included: one-year mortality, hospital readmission for any reason or emergency department presentation for any reason at one-year and a composite of death, myocardial infarction or cardiac arrest at one-year.

## Statistical analysis

Descriptive analysis was undertaken to demonstrate differences between HFRS risk categories. Categorical variables were summarized using frequency and column percentage and continuous variables were described using mean and standard deviation. Non-normal distributed data is presented as median and inter-quartile ranges. Unadjusted odds ratio (OR) and 95% confidence intervals (CI) were derived from univariate logistic regression. Adjusted odds ratio (OR) and 95% confidence intervals (CI) were derived from multivariable logistic regression adjusted for age, sex, categorized number of preadmissions, categorized RCRI score, troponin, b-natriuretic peptide and the 17 components of Charlson Comorbidity Index (CCI). Stepwise variable selection (with p = 0.05 as enter and stay criterion) was adopted to get rid of redundant variables.

The incremental predictive power of including HFRS for outcomes was assessed using area under the Receiver Operating Characteristic (AUROC) curve and a net reclassification improvement (NRI) analysis. For both analyses the study cohort was randomly and evenly split into a training and a validation dataset. The following predictive models were built on the training and tested on the validation datasets for assessing predictive power of including HFRS. One is the gold standard model (reference) derived from a multivariable logistic regression which included the following candidate variables: age, sex, components of the CCI, RCRI as a categorical variable, categorized pre-operative troponin and pre-operative BNP values. The second is the proposed model which included the addition of HFRS as a categorical variable. Stepwise variable selection was used to include informative predictors into the final predictive models with p = 0.1 as enter and stay criterion.

NRI generates three values of relevance to be considered. The first is the NRI for events, which examines the proportion of patients with a correctly predicted observed outcome. An improvement in the events NRI represents a reduction in type I error (false positives), whereas

a reduction in the events NRI represents an increase in type I error. NRI for non-events examines the proportion of patients who were correctly predicted to not experience an outcome. An improvement in the non-events NRI represents a reduction in type II error (false negatives), whereas a reduction in the non-events NRI represents an increase in type II error. The overall NRI is the sum of the events NRI and the non-events NRI.

All Statistical analysis was performed using Statistical Analysis System (SAS) Enterprise Guide 7.1 (Cary, NC, USA) and R (R Core Team (2021)).

### Ethics

This study complies with the Declaration of Helsinki and was approved by the University of Alberta Health Research Ethics Board (Pro00081737) with a waiver of informed patient consent due to the minimal risk and being infeasible to contact all patients.

## Results

### Patient demographics

During the study period, 798,918 non-cardiac surgical admissions were identified, of which 86,048 were excluded due to non-index surgical procedure and 62 were excluded due to a second surgical procedure during the same admission. The final cohort consisted of 712,808 patients. The average age of patients was 53·4 +/- 22·4 years and 55·1% were female (Table 1). The cohort included 300,877 (42·2%) patients undergoing minor surgeries, 227,183 (31·9%) abdominal surgeries, 210,536 (29·5%) orthopedic surgeries, 85,562 (12·0%) pelvic surgeries, 7,833 (1·1%) thoracic surgeries and 7,861 (1·1%) vascular surgeries. Details of specific surgery types can be found in S4 Table. The majority of patients had an RCRI score of 0 (54·7%) and the mean CCI was 0·5 +/- 0·8. Using the HFRS, 86·3% (n = 614,921) were considered low risk, 10·7% (n = 76,136) were considered intermediate risk and 3·1% (n = 21,751) were considered high risk for frailty.

### Primary outcome

The primary outcome composite of death, myocardial infarction or cardiac arrest at 30-days occurred in 4636 patients, ranging from 0·4% in the HFRS low risk group to 2·8% in the HFRS high risk group (Table 2 and Fig 1). In the adjusted model, an intermediate and high HFRS score was associated with an increased risk of the primary outcomes (OR 1·61; 1·5–1·74 and 1·55, 1·38–1·73). The breakdown of major adverse cardiovascular events (MACE) categorised by RCRI and HFRS can be seen in Fig 2 and S5 Table.

### Short-term outcomes

A higher HFRS score was associated with a prolonged length of stay, ranging from 27·7% in the low HFRS group to 67·6% in the high HFRS group (Fig 1). HFRS was strongly associated with increased odds of in-hospital mortality (Fig 1), with the intermediate HFRS group demonstrating an adjusted OR of 6·99 (6·63–7·37) and the high HFRS group demonstrating an aOR of 9·67 (8·99–10·39). Odds of 30-day mortality (Fig 1) and 30-day readmission/ED visit (Fig 1) were similarly higher in the intermediate and high risk HFRS groups (Table 2).

### Long-term outcomes

The composite of death, myocardial infarction or cardiac arrest at one-year occurred in 2·2% of patients in low risk HFRS group, 9·6% (aOR 1·55; 1·49–1·61) in the intermediate HFRS group and 13·1% (aOR 1·58; 1·49–1·67) in the high HFRS group. One-year mortality occurred

**Table 1. Patient characteristics stratified by hospital frailty risk score.**

| | Low risk (<5) | Mediate risk (5–15) | High risk (>15) | Overall | p-value |
|---|---|---|---|---|---|
| N | 614921 | 76136 | 21751 | 712808 | |
| **Age (years)** | | | | | <.0001 |
| mean (SD) | 51.0 (22.0) | 67.0 (19.6) | 73.5 (16.5) | 53.4 (22.4) | |
| median (IQR) | 54.0 (37.0–67.0) | 71.0 (57.0–82.0) | 78.0 (65.0–85.0) | 57.0 (39.0–70.0) | |
| **Female n (%)** | 342701 (55.7%) | 37867 (49.7%) | 11933 (54.9%) | 392501 (55.1%) | <.0001 |
| **HFRS Score** | | | | | <.0001 |
| mean (SD) | 0.6 (1.2) | 8.7 (2.8) | 21.1 (5.5) | 2.1 (4.5) | |
| median (IQR) | 0.0 (0.0–0.9) | 8.1 (6.3–10.7) | 19.5 (16.9–23.6) | 0.0 (0.0–2.1) | |
| **CCI Score** | | | | | <.0001 |
| mean (SD) | 0.4 (0.7) | 1.2 (1.1) | 1.6 (1.2) | 0.5 (0.8) | |
| median (IQR) | 0.0 (0.0–1.0) | 1.0 (0.0–2.0) | 1.0 (1.0–2.0) | 0.0 (0.0–1.0) | |
| **CCI Score, n (%)** | | | | | <.0001 |
| = 0 | 440721 (71.7%) | 26256 (34.5%) | 4428 (20.4%) | 471405 (66.1%) | |
| = 1 | 122414 (19.9%) | 24897 (32.7%) | 7309 (33.6%) | 154620 (21.7%) | |
| ≥ 2 | 51786 (8.4%) | 24983 (32.8%) | 10014 (46.0%) | 86783 (12.2%) | |
| **Number of admissions in preceding 2 years, n (%)** | | | | | <.0001 |
| = 1 | 483436 (78.6%) | 28845 (37.9%) | 3883 (17.9%) | 516164 (72.4%) | |
| = 2 | 96942 (15.8%) | 21367 (28.1%) | 5089 (23.4%) | 123398 (17.3%) | |
| ≥ 3 | 34543 (5.6%) | 25924 (34.0%) | 12779 (58.8%) | 73246 (10.3%) | |
| **RCRI Score, n (%)** | | | | | <.0001 |
| = 0 | 348747 (56.7%) | 34092 (44.8%) | 7024 (32.3%) | 389863 (54.7%) | |
| = 1 | 240382 (39.1%) | 24480 (32.2%) | 6972 (32.1%) | 271834 (38.1%) | |
| = 2 | 4129 (0.7%) | 4472 (5.9%) | 2218 (10.2%) | 10819 (1.5%) | |
| ≥ 3 | 21663 (3.5%) | 13092 (17.2%) | 5537 (25.5%) | 40292 (5.7%) | |
| **Troponin, n (%)** | | | | | <.0001 |
| No test | 561644 (91.3%) | 51628 (67.8%) | 13964 (64.2%) | 627236 (88.0%) | |
| Normal | 49374 (8.0%) | 20447 (26.9%) | 6513 (29.9%) | 76334 (10.7%) | |
| High | 3903 (0.6%) | 4061 (5.3%) | 1274 (5.9%) | 9238 (1.3%) | |
| **BNP, n (%)** | | | | | <.0001 |
| No test | 561644 (91.3%) | 51628 (67.8%) | 13964 (64.2%) | 627236 (88.0%) | |
| Normal | 51017 (8.3%) | 22759 (29.9%) | 7189 (33.1%) | 80965 (11.4%) | |
| High | 2260 (0.4%) | 1749 (2.3%) | 598 (2.7%) | 4607 (0.6%) | |
| **Surgery Type, n (%)** | | | | | |
| Vascular Surgery | 6423 (1.0%) | 1263 (1.7%) | 175 (0.8%) | 7861 (1.1%) | <.0001 |
| Abdominal Surgery | 212302 (34.5%) | 12456 (16.4%) | 2425 (11.1%) | 227183 (31.9%) | <.0001 |
| Thoracic Surgery | 7274 (1.2%) | 515 (0.7%) | 44 (0.2%) | 7833 (1.1%) | <.0001 |
| Pelvic Surgery | 83073 (13.5%) | 2173 (2.9%) | 316 (1.5%) | 85562 (12.0%) | <.0001 |
| Orthopedic Surgery | 180542 (29.4%) | 22407 (29.4%) | 7587 (34.9%) | 210536 (29.5%) | <.0001 |
| Minor Surgery | 239209 (38.9%) | 47964 (63.0%) | 13704 (63.0%) | 300877 (42.2%) | <.0001 |

Abbreviations: HFRS–Hospital frailty risk score; CCI–Charlson comorbidity index; FRU–fracture, radius and ulna; FTF—fracture, tibia and fibular.

Note: p values are from non-parametric Kruskal-Wallis test for continuous variables and Chi-square test for categorical variables.

in 1·4% of patients in low risk HFRS group, 6·5% (aOR 1·77; 1·69–1·85) in the intermediate HFRS group and 9·2% (aOR 1·93; 1·8–2·06) in the high HFRS group. One-year readmission or ED visit occurred in 38·6% of patients in low risk HFRS group, 62·2% (aOR 1·56; 1·53–1·58) in the intermediate HFRS group and 64·6% (aOR 1·42; 1·37–1·47) in the high HFRS group.

**Table 2. Outcomes categorized by hospital frailty risk score.**

| | Low risk (<5) | Mediate risk (5–15) | High risk (>15) | p-value |
|---|---|---|---|---|
| **Prolonged hospital stay, n (%)** | | | | |
| Raw rate, n (%) | 170272 (27.7%) | 42406 (55.7%) | 14710 (67.6%) | <.0001 |
| Unadjusted OR (95%CI) | 1 (reference) | 3.28 (3.23, 3.33) | 5.46 (5.3, 5.62) | <.0001 |
| Fully Adjusted OR (95%CI) | 1 (reference) | 2.08 (2.04, 2.11) | 2.59 (2.51, 2.68) | <.0001 |
| **In-hospital mortality, n (%)** | | | | |
| Raw rate, n (%) | 3220 (0.5%) | 5231 (6.9%) | 2104 (9.7%) | <.0001 |
| Unadjusted OR (95%CI) | 1 (reference) | 14.01 (13.4, 14.65) | 20.34 (19.22, 21.53) | <.0001 |
| Fully Adjusted OR (95%CI) | 1 (reference) | 6.99 (6.63, 7.37) | 9.67 (8.99, 10.39) | <.0001 |
| **30-day ED visit or readmission, n (%)** | | | | |
| Raw rate, n (%) | 87531 (14.3%) | 16590 (23.4%) | 4707 (24.0%) | <.0001 |
| Unadjusted OR (95%CI) | 1 (reference) | 1.83 (1.8, 1.86) | 1.89 (1.82, 1.95) | <.0001 |
| Fully Adjusted OR (95%CI) | 1 (reference) | 1.27 (1.25, 1.3) | 1.17 (1.12, 1.22) | <.0001 |
| **30-day mortality, n (%)** | | | | |
| Raw rate, n (%) | 918 (0.2%) | 574 (0.8%) | 211 (1.1%) | <.0001 |
| Unadjusted OR (95%CI) | 1 (reference) | 5.43 (4.89, 6.03) | 7.22 (6.21, 8.39) | <.0001 |
| Fully Adjusted OR (95%CI) | 1 (reference) | 2.26 (2.01, 2.54) | 2.67 (2.26, 3.16) | <.0001 |
| **30-day composite of death, MI or cardiac arrest, n (%)** | | | | |
| Raw rate, n (%) | 2546 (0.4%) | 1533 (2.2%) | 557 (2.8%) | <.0001 |
| Unadjusted OR (95%CI) | 1 (reference) | 5.29 (4.96, 5.64) | 6.98 (6.36, 7.66) | <.0001 |
| Fully Adjusted OR (95%CI) | 1 (reference) | 1.61 (1.5, 1.74) | 1.55 (1.38, 1.73) | <.0001 |
| **1-year ED Visit or readmission, n (%)** | | | | |
| Raw rate, n (%) | 235918 (38.6%) | 44073 (62.2%) | 12701 (64.6%) | <.0001 |
| Unadjusted OR (95%CI) | 1 (reference) | 2.62 (2.57, 2.66) | 2.91 (2.83, 3) | <.0001 |
| Fully Adjusted OR (95%CI) | 1 (reference) | 1.56 (1.53, 1.58) | 1.42 (1.37, 1.47) | <.0001 |
| **1-year mortality, n (%)** | | | | |
| Raw rate, n (%) | 8616 (1.4%) | 4624 (6.5%) | 1809 (9.2%) | <.0001 |
| Unadjusted OR (95%CI) | 1 (reference) | 4.88 (4.71, 5.07) | 7.1 (6.73, 7.48) | <.0001 |
| Fully Adjusted OR (95%CI) | 1 (reference) | 1.77 (1.69, 1.85) | 1.93 (1.8, 2.06) | <.0001 |
| **1-year composite of death, MI or cardiac arrest, n (%)** | | | | |
| Raw rate, n (%) | 13305 (2.2%) | 6788 (9.6%) | 2573 (13.1%) | <.0001 |
| Unadjusted OR (95%CI) | 1 (reference) | 4.76 (4.62, 4.91) | 6.78 (6.48, 7.09) | <.0001 |
| Fully Adjusted OR (95%CI) | 1 (reference) | 1.55 (1.49, 1.61) | 1.58 (1.49, 1.67) | <.0001 |

• Unadjusted OR: from univariate logistic regression.

• Fully adjusted OR: from multivariable logistic regression adjusted for age, sex, categorized number of preadmissions, RCRI score, troponin, BNP and the 17 components of CCI.

### Incremental benefit of HFRS

The NRI analysis demonstrated that for all outcomes, the addition of HFRS improved overall predictions (Fig 3 and S6 Table). For the primary composite outcomes of death, hospitalisation for MI or cardiac arrest at 30-days the overall NRI, NRI for events and NRI for non-events was 0.133, -0.175 and 0.288 respectively. When examining NRI events and NRI non-events separately, we observe that the inclusion of HFRS consistently improved the ability to identify non-events, but was generally associated with an increase in false positives. The only exception was in-hospital mortality, where the HFRS improved the predictive abilities for both events (0.320) and non-events (0.634). The AUROC analysis demonstrated similar results (S7 Table), with

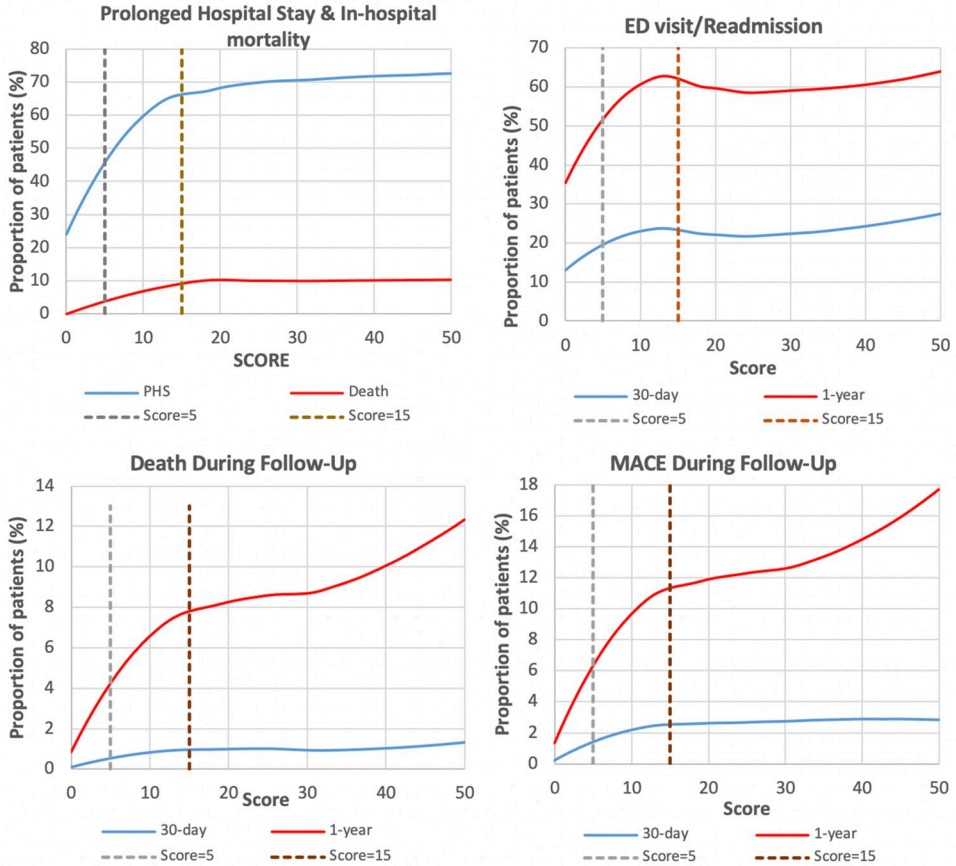

**Fig 1. Proportion of patients with adverse event by hospital frailty risk score.** Spectrum of adverse events across the HFRS spectrum. Abbreviations: PHS–prolonged hospital stay; HFRS–Hospital Frailty Risk Score.

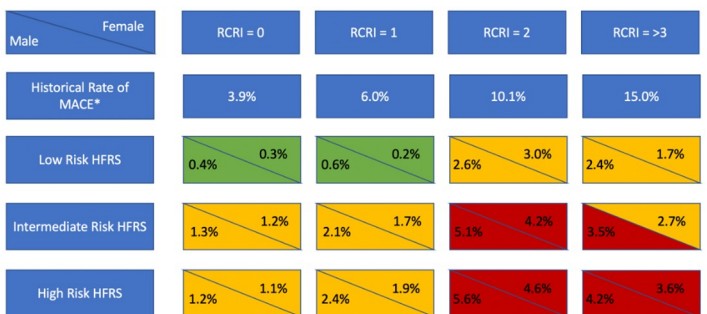

**Fig 2. Rate of 30-day major adverse cardiovascular events categorised by RCRI and HFRS.** * Historical rate of MACE (composite of death, myocardial infarction or cardiac arrest at 30-days) from the 2017 Canadian Cardiovascular Society Guidelines of Perioperative Cardiac Risk Assessment and Management for Patients Who Undergo Noncardiac Surgery. Green– 30-day MACE <1%, Yellow– 30-day MACE 1–3%, Red– 30-day MACE >3%; Abbreviations: RCRI–Revised Cardiac Risk Index; HFRS–Hospital Frailty Risk Score; MACE–major adverse cardiovascular events.

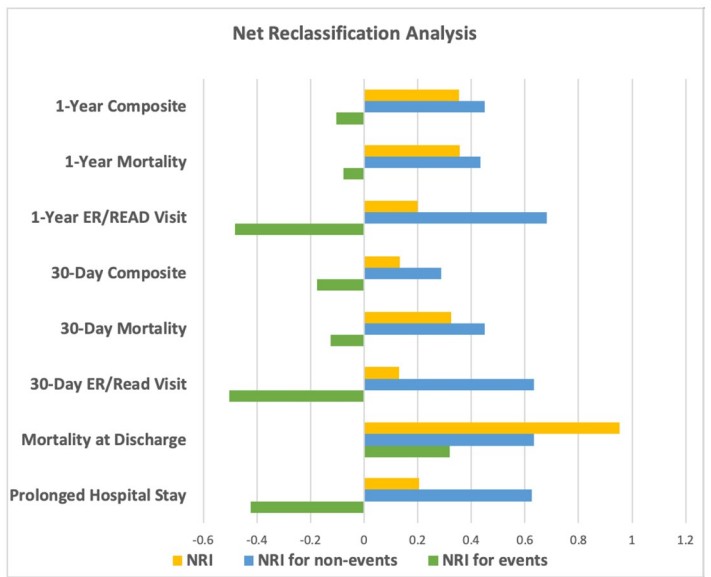

**Fig 3. Visual representation of net reclassification improvement analysis.** Negative NRI is interpreted as an increase in error and a positive NRI is interpreted as a decrease in error. Yellow–Overall NRI, Blue–NRI for non-events, Green–NRI for events.

statistically significant improvements for all outcomes with the greatest AUROC improvement observed for in-hospital mortality (3.0%).

## Discussion

In a large, generalizable cohort of patients undergoing non-cardiac surgery, frailty identified by the Hospital Frailty Risk Score was associated with adverse short- and long-term outcomes. The hospital frailty risk score provides incremental prognostic information to traditional RCRI risk estimation. For example, for patients with an RCRI of one, the risk of 30-day death, MI and cardiac arrest ranges from 0.35% in the low frailty risk group to 2.12% in the high-risk frailty group.

Since the development and validation of the hospital frailty risk score in the acute care setting [9], the association between HFRS score and adverse events has been replicated in numerous in-patient [11–14] and procedural settings [15–17]. In the peri-operative setting, higher HFRS scores have been associated with adverse events in patients undergoing spinal surgery [18], joint arthroplasty [19], vascular surgery [20] and cirrhotic patients undergoing surgery [21]. In a generalizable cohort of 487,197 patients over the age of 50 undergoing surgery Harvey et al. demonstrated that a high HFRS was associated with prolonged length of stay, 30-day mortality and 28-day readmission [22]. However, the addition of HRFS to CCI did not significantly improve model performance. In contrast, in our analysis, after adjusting for RCRI and CCI, HFRS remained a significant predictor of both short and long-term outcomes. The difference is likely the result of variations in coding practices—in our analysis, we are able to capture ICD codes from all in-patient, emergency department and outpatient encounters. In comparison, in the analysis by Harvey et al. the CCI and HFRS were not calculated in 17% and 32% of individuals due to a lack of hospitalisation in the previous two and five years respectively [22]. This highlights the potential importance of a broad, standardized approach to collecting

administrative data for HFRS calculation and the implications of coding depth for accurate HFRS calculation [23].

Overall, frailty defined using a variety of assessment tools has been demonstrated to be associated with adverse peri-operative outcomes [24, 25]. It is important to consider that frailty takes many shapes and form, and as a result, a multidimensional approach to assessment that assesses physical, mental, nutritional, and socioeconomic factors is the ideal. This approach has been demonstrated to be valid and correlated with adverse events in community-dwelling older adults and in patients with heart failure [26, 27]. However, HFRS provides a unique opportunity for implementation in electronic medical record systems, due to its ability to be tabulated by readily available administrative data without the need for the manual application of frailty assessment. The availability of this point-of-care estimate of frailty provides valuable information to physicians and surgeons, allowing for a more nuanced discussion of peri-operative risk. The HFRS may also be used to identify a high-risk patient population that would warrant dedicated frailty assessment and consideration for enrollment in pre-operative prehabilitation interventions that have been shown to be beneficial in surgical setting [28, 29]. Based on our NRI analysis, the greatest strength of the HFRS is its ability to identify patients that would traditionally be deemed high risk based on their RCRI score and reclassify them to a lower risk based on the absence of frailty metrics included in the HFRS. Further studies are warranted to examine whether the HFRS can provide clinically actionable information to surgeons and pre-operative assessors, identify appropriate candidates for prehabilitation and be integrated into routine clinical practice and be utilised to judiciously utilise peri operative testing. In addition, further analysis is required to determine whether the impact of frailty as measured by the HFRS varies based on the spectrum of surgical risk for any given procedure.

Limitations of an ICD-based frailty assessment revolve around the accuracy and depth of coding to accurately define frailty. In addition, ICD coding does not allow for severity of conditions to be considered. As previously mentioned, administrative data is not primarily intended for research purposes, further increasing the risk of incorrect coding or missing data bias. In our analysis, we retrospectively utilised this administrative data, however it is unclear it can be prospectively used in the future to be utilised in direct patient care. Further studies are required to assess the feasibility of implementation of HFRS into existing electronic medical records and its impact on peri-operative decision making. In large sample size populations, such as in our study, even small effect sizes may demonstrate statistical significance. Clinical relevance of such small sizes should be taken into consideration when interpreting these statistically significant results. In the present study, risk of MI, cardiac arrest and death was significantly lower compared to pooled event rates from previous observational studies [30], however these prior studies were frequently small, included a small proportion of surgical procedures and were conducted more than a decade ago.

In conclusion, in a large, inclusive population of all adults undergoing non-cardiac surgery, the HFRS provides important short- and long-term prognostic information above and beyond traditional peri-operative risk stratification.

## Supporting information

**S1 Table. Procedure codes for non-cardiac surgery.**
(DOCX)

**S2 Table. Hospital frailty risk score weights by ICD code.**
(DOCX)

**S3 Table. Estimated length of stay by surgery type.**
(DOCX)

**S4 Table. Breakdown of HFRS by surgery.**
(DOCX)

**S5 Table. Rate of major cardiac events (death, myocardial infarction or cardiac arrest within 30-days of discharge) stratified by RCRI, HFRS and sex.**
(DOCX)

**S6 Table. Net reclassification improvement analysis.**
(DOCX)

**S7 Table. Area under the receiver operating characteristic analysis.**
(DOCX)

## Author Contributions

**Conceptualization:** Pishoy Gouda, Michelle M. Graham.

**Data curation:** Xiaoming Wang, Erik Youngson.

**Formal analysis:** Xiaoming Wang, Erik Youngson.

**Investigation:** Pishoy Gouda, Xiaoming Wang, Mamas A. Mamas, Michelle M. Graham.

**Methodology:** Pishoy Gouda, Erik Youngson, Michael McGillion, Michelle M. Graham.

**Project administration:** Michelle M. Graham.

**Resources:** Michelle M. Graham.

**Supervision:** Michelle M. Graham.

**Validation:** Xiaoming Wang, Erik Youngson.

**Visualization:** Michael McGillion, Michelle M. Graham.

**Writing – original draft:** Pishoy Gouda, Michelle M. Graham.

**Writing – review & editing:** Pishoy Gouda, Michael McGillion, Mamas A. Mamas, Michelle M. Graham.

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
