## [Decision Letter · Decision Letter 0]

15 Nov 2021

PONE-D-21-33039Beyond the Revised Cardiac Risk Index: Validation of the Hospital Frailty Risk Score in Non-Cardiac SurgeryPLOS ONE

Dear Dr. GRAHAM,

Thank you for submitting your manuscript to PLOS ONE. After careful consideration, we feel that it has merit but does not fully meet PLOS ONE’s publication criteria as it currently stands. Therefore, we invite you to submit a revised version of the manuscript that addresses the points raised during the review process.

We look forward to receiving your revised manuscript.

Kind regards,

Pasquale Abete

Academic Editor

PLOS ONE

Journal Requirements:

Additional Editor Comments:

Accordng to Reviewers'decision I suggest a minor revision of the manuscript.

Reviewers' comments:

Reviewer's Responses to Questions

**Comments to the Author**

1. Is the manuscript technically sound, and do the data support the conclusions?

Reviewer #1: Yes

Reviewer #2: Yes

2. Has the statistical analysis been performed appropriately and rigorously? 

Reviewer #1: Yes

Reviewer #2: Yes

3. Have the authors made all data underlying the findings in their manuscript fully available?

Reviewer #1: Yes

Reviewer #2: No

4. Is the manuscript presented in an intelligible fashion and written in standard English?

Reviewer #1: Yes

Reviewer #2: Yes

5. Review Comments to the Author

Reviewer #1: The Authors evaluated the incremental prognostic utility of the HFRS in a generalizable surgical population. Using linked administrative databases, a huge (712,808 non-cardiac surgeries) retrospective cohort of patients admitted over 10 years in Alberta, Canada was created. The primary outcome was a composite of death, myocardial infarction or cardiac arrest at 30-days. Multivariable logistic regression was undertaken to assess the impact of HFRS on outcomes after adjusting for age, sex, components of the Charlson Comorbidity Index (CCI), Revised Cardiac Risk Index (RCRI) and perioperative biomarkers. Using the HFRS, 86.3% were considered low risk, 10·7% were considered intermediate risk and 3·1% were considered high risk for frailty. Intermediate and high HFRS scores were associated with increased risk of the primary outcome with an adjusted odds ratio of 1·61 (95% CI 1·50-1.74) and 1·55 (95% CI 1·38-1·73). Intermediate and high HFRS were also associated with increased adjusted odds of prolonged hospital stay, in-hospital mortality, and 1-year mortality. I found the manuscript of great interest. The study is well conducted. Results support conclusion. I have no suggestions to improve your excellent manuscript.

Reviewer #2: The authors used the Hospital Frailty Risk Score (HFRS), recently described frailty assessment tool that harnesses administrative data, to examine the incremental prognostic utility of the HFRS in a generalizable surgical population. Methods Using linked administrative databases, a retrospective cohort of patients admitted for non-cardiac surgery between October 1st, 2008 and September 30th, 2019 in Alberta, Canada was analyzed. Our primary outcome was a composite of death, myocardial infarction or cardiac arrest at 30-days. Multivariable logistic regression was undertaken to assess the impact of HFRS on outcomes after adjusting for age, sex, components of the Charlson Comorbidity Index (CCI), Revised Cardiac Risk Index (RCRI) and perioperative biomarkers. The final cohort consisted of 712,808 non-cardiac surgeries, of which 55•1% were female and the average age was 53•4 +/- 22•4 years. Using the HFRS, 86.3% were considered low risk, 10•7% were considered intermediate risk and 3•1% were considered high risk for frailty. Intermediate and high HFRS scores were associated with increased risk of the primary outcome with an adjusted odds ratio of 1•61 (95% CI 1•50-1.74) and 1•55 (95% CI 1•38-1•73). Intermediate and high HFRS were also associated with increased adjusted odds of prolonged hospital stay, in-hospital mortality, and 1-year mortality.

The manuscript is interesting. However I suggest adding a section bout the limitations of the study underlying the use of administrative data and the limits of a “retrospective study”. In addition you should stress the need of multidimensional approach to frailty (see and discuss Abete P et al. The Italian version of the "frailty index" based on deficits in health: a validation study. Aging Clin Exp Res. 2017 Oct;29(5):913-926) and, more importantly in heart disease patients (see and discuss Testa G et al. Physical vs. multidimensional frailty in older adults with and without heart failure. ESC Heart Fail. 2020 Jun;7(3):1371-1380).

6. PLOS authors have the option to publish the peer review history of their article (what does this mean?). If published, this will include your full peer review and any attached files.

Reviewer #1: No

Reviewer #2: No

---

## [Author Response · Author response to Decision Letter 0]

21 Nov 2021

Manuscript title: Beyond the Revised Cardiac Risk Index: Validation of the Hospital Frailty Risk Score in Non-Cardiac Surgery

Manuscript ID: PONE-D-21-33039

Dear Dr. Pasquale Abete,

We thank the editor and external reviewers for their time and insightful comments, and recognize the manuscript is substantially improved as a result. Please find below a detailed response to each of comments.

Journal Requirements:

Comment # 1 - Please ensure that your manuscript meets PLOS ONE's style requirements, including those for file naming. 

Response to comment # 1 – We have gone through the PLOS ONE style requirements and style templates and have made the necessary changes as required.

Comment # 2 - Please review your reference list to ensure that it is complete and correct. If you have cited papers that have been retracted, please include the rationale for doing so in the manuscript text, or remove these references and replace them with relevant current references. Any changes to the reference list should be mentioned in the rebuttal letter that accompanies your revised manuscript. If you need to cite a retracted article, indicate the article’s retracted status in the References list and also include a citation and full reference for the retraction notice.

Response to comment # 2 – We have gone through the reference list and ensured they are complete and correctly formatted. 

Editor Comments:

Comment # 1 - According to Reviewers' decision I suggest a minor revision of the manuscript.

Response to comment # 1 – Thank you for your assistance in the review process. We have gone through the reviewer comments and made the requested changes. Please see below for details.

Reviewer #1: 

Comment # 1 - The Authors evaluated the incremental prognostic utility of the HFRS in a generalizable surgical population. Using linked administrative databases, a huge (712,808 non-cardiac surgeries) retrospective cohort of patients admitted over 10 years in Alberta, Canada was created. The primary outcome was a composite of death, myocardial infarction or cardiac arrest at 30-days. Multivariable logistic regression was undertaken to assess the impact of HFRS on outcomes after adjusting for age, sex, components of the Charlson Comorbidity Index (CCI), Revised Cardiac Risk Index (RCRI) and perioperative biomarkers. Using the HFRS, 86.3% were considered low risk, 10·7% were considered intermediate risk and 3·1% were considered high risk for frailty. Intermediate and high HFRS scores were associated with increased risk of the primary outcome with an adjusted odds ratio of 1·61 (95% CI 1·50-1.74) and 1·55 (95% CI 1·38-1·73). Intermediate and high HFRS were also associated with increased adjusted odds of prolonged hospital stay, in-hospital mortality, and 1-year mortality. I found the manuscript of great interest. The study is well conducted. Results support conclusion. I have no suggestions to improve your excellent manuscript.

Response to comment # 1 – Thank you for your time and consideration in the peer review process. 

Reviewer #2: 

Comment # 1 - The authors used the Hospital Frailty Risk Score (HFRS), recently described frailty assessment tool that harnesses administrative data, to examine the incremental prognostic utility of the HFRS in a generalizable surgical population. Methods Using linked administrative databases, a retrospective cohort of patients admitted for non-cardiac surgery between October 1st, 2008 and September 30th, 2019 in Alberta, Canada was analyzed. Our primary outcome was a composite of death, myocardial infarction or cardiac arrest at 30-days. Multivariable logistic regression was undertaken to assess the impact of HFRS on outcomes after adjusting for age, sex, components of the Charlson Comorbidity Index (CCI), Revised Cardiac Risk Index (RCRI) and perioperative biomarkers. The final cohort consisted of 712,808 non-cardiac surgeries, of which 55•1% were female and the average age was 53•4 +/- 22•4 years. Using the HFRS, 86.3% were considered low risk, 10•7% were considered intermediate risk and 3•1% were considered high risk for frailty. Intermediate and high HFRS scores were associated with increased risk of the primary outcome with an adjusted odds ratio of 1•61 (95% CI 1•50-1.74) and 1•55 (95% CI 1•38-1•73). Intermediate and high HFRS were also associated with increased adjusted odds of prolonged hospital stay, in-hospital mortality, and 1-year mortality.

Response to comment # 1 – Thank you for your time and consideration in the peer review process. 

Comment # 2 - The manuscript is interesting; however, I suggest adding a section about the limitations of the study underlying the use of administrative data and the limits of a “retrospective study”. 

Response to comment # 2 – In the limitations section (page 16, line 270-279) we have added additional details regarding the limitations of administrative data and the retrospective use of this data. “Limitations of an ICD-based frailty assessment revolve around the accuracy and depth of coding to accurately define frailty. In addition, ICD coding does not allow for severity of conditions to be considered. As previously mentioned, administrative data is not primarily intended for research purposes, further increasing the risk of incorrect coding or missing data bias. In our analysis, we retrospectively utilised this administrative data, however it is unclear it can be prospectively used in the future to be utilised in direct patient care.”.

Comment # 3 - In addition you should stress the need of multidimensional approach to frailty (see and discuss Abete P et al. The Italian version of the "frailty index" based on deficits in health: a validation study. Aging Clin Exp Res. 2017 Oct;29(5):913-926) and, more importantly in heart disease patients (see and discuss Testa G et al. Physical vs. multidimensional frailty in older adults with and without heart failure. ESC Heart Fail. 2020 Jun;7(3):1371-1380).

Response to comment # 3 – Thank you for your suggestion. We have expanded our discussion to highlight the importance of a multidimensional approach to the assessment of frailty and its use in community dwelling older adults and in patients with heart failure citing Abete et al. and Testa et al. This has been added to page 15, line 253-257, “It is important to consider that frailty takes many shapes and form, and as a result, a multidimensional approach to assessment that assesses physical, mental, nutritional, and socioeconomic factors is the ideal. This approach has been demonstrated to be valid and correlated with adverse events in community-dwelling older adults and in patients with heart failure [26, 27].”.

---

## [Decision Letter · Decision Letter 1]

23 Dec 2021

Beyond the Revised Cardiac Risk Index: Validation of the Hospital Frailty Risk Score in Non-Cardiac Surgery

PONE-D-21-33039R1

Dear Dr. GRAHAM,

We’re pleased to inform you that your manuscript has been judged scientifically suitable for publication and will be formally accepted for publication once it meets all outstanding technical requirements.

Kind regards,

Pasquale Abete

Academic Editor

PLOS ONE

Additional Editor Comments (optional):

No more comments.

Reviewers' comments:

Reviewer's Responses to Questions

**Comments to the Author**

1. If the authors have adequately addressed your comments raised in a previous round of review and you feel that this manuscript is now acceptable for publication, you may indicate that here to bypass the “Comments to the Author” section, enter your conflict of interest statement in the “Confidential to Editor” section, and submit your "Accept" recommendation.

Reviewer #2: All comments have been addressed

2. Is the manuscript technically sound, and do the data support the conclusions?

Reviewer #2: Yes

3. Has the statistical analysis been performed appropriately and rigorously? 

Reviewer #2: Yes

4. Have the authors made all data underlying the findings in their manuscript fully available?

Reviewer #2: Yes

5. Is the manuscript presented in an intelligible fashion and written in standard English?

Reviewer #2: Yes

6. Review Comments to the Author

Reviewer #2: The manuscript is really improved, All questions have been addressed. No further comments are needed.

7. PLOS authors have the option to publish the peer review history of their article (what does this mean?). If published, this will include your full peer review and any attached files.

Reviewer #2: No

---

## [Editor Report · Acceptance letter]

26 Dec 2021

PONE-D-21-33039R1 

Beyond the Revised Cardiac Risk Index: Validation of the Hospital Frailty Risk Score in non-cardiac surgery 

Dear Dr. Graham:

I'm pleased to inform you that your manuscript has been deemed suitable for publication in PLOS ONE. Congratulations! Your manuscript is now with our production department. 

Kind regards, 

on behalf of

Prof. Pasquale Abete 

Academic Editor

PLOS ONE